# SMALL-GAN: SPEEDING UP GAN TRAINING USING CORE-SETS

## ABSTRACT

Recent work by Brock et al. (2018) suggests that Generative Adversarial Networks (GANs) benefit disproportionately from large mini-batch sizes. Unfortunately, using large batches is slow and expensive on conventional hardware. Thus, it would be nice if we could generate batches that were *effectively large* though actually small. In this work, we propose a method to do this, inspired by the use of Coreset-selection in active learning. When training a GAN, we draw a large batch of samples from the prior and then compress that batch using Coreset-selection. To create effectively large batches of 'real' images, we create a cached dataset of Inception activations of each training image, randomly project them down to a smaller dimension, and then use Coreset-selection on those projected activations at training time. We conduct experiments showing that this technique substantially reduces training time and memory usage for modern GAN variants, that it reduces the fraction of dropped modes in a synthetic dataset, and that it allows GANs to reach a new state of the art in anomaly detection.

## 1 INTRODUCTION

Generative Adversarial Networks (GANs) (Goodfellow et al., 2014) have become a popular research topic. Arguably the most impressive results have been in image synthesis (Brock et al., 2018; Salimans et al., 2018; Miyato et al., 2018; Zhang et al., 2018; 2017), but they have also been applied fruitfully to text generation (Fedus et al., 2018; Guo et al., 2018), domain transfer learning (Zhu et al., 2017; Zhang et al., 2017; Isola et al., 2017), and various other tasks (Xian et al., 2018; Ledig et al., 2017; Zhu & Bento, 2017).

Recently, Brock et al. (2018) substantially improved the results of Zhang et al. (2018) by using very large mini-batches during training. The effect of large mini-batches in the context of deep learning is well-studied (Smith et al., 2017; Goyal et al., 2017; Keskar et al., 2016; Shallue et al., 2018) and general consensus is that they can be helpful in many circumstances, but the results of Brock et al. (2018) suggest that GANs benefit disproportionately from large batches. In fact, Table 1 of Brock et al. (2018) shows that for the Frechet Inception Distance (FID) metric (Heusel et al., 2017) on the ImageNet dataset, scores can be improved from 18.65 to 12.39 simply by making the batch eight times larger.

Unfortunately, increasing the batch size in this manner is not always possible since it increases the computational resources required to train these models – often beyond the reach of conventional hardware. The experiments from the BigGAN paper require a full 'TPU Pod'. The 'unofficial' open source release of BigGAN works around this by accumulating gradients across 8 different V100 GPUs and only taking an optimizer step every 8 gradient accumulation steps. Future research on GANs would be much easier if we could have the gains from large batches without these pain points. In this paper, we take steps toward accomplishing that goal by proposing a technique that allows for *mimicking* large batches without the computational costs of actually using large batches.

In this work, we use Core-set selection (Agarwal et al., 2005) to sub-sample a large batch to produce a smaller batch. The large batches are then discarded, and the sub-sampled, smaller, batches are used to train the GAN. Informally, this procedure yields small batches with 'coverage' similar to that of the large batch – in particular the small batch tries to 'cover' all the same modes as are covered in the large batch. This technique yields many of the benefits of having large batches with much less computational overhead. Moreover, it is generic, and so can be applied to nearly all GAN variants.

Our contributions can be summarized as follows:

- We introduce a simple, computationally cheap method to increase the 'effective batch size' of GANs, which can be applied to any GAN variant.
- We conduct experiments on the CIFAR and LSUN datasets showing that our method can substantially improve FID across different GAN architectures given a fixed batch size.
- We use our method to improve the performance of the technique from Kumar et al. (2019), resulting in state-of-the-art performance at GAN-based anomaly detection.

## 2 BACKGROUND AND NOTATION

**Generative Adversarial Networks** A Generative Adversarial Network (or GAN) is a system of two neural networks trained 'adversarially'. The generator, $G$, takes as input samples from a prior $z \sim p(z)$ and outputs the learned distribution, $G(z)$. The discriminator, $D$, receives as input both the training examples, $X$, and the synthesized samples, $G(z)$, and outputs a distribution $D(.)$ over the possible sample source. The discriminator is then trained to maximize the following objective:

$$\mathcal{L}_D = -\mathbb{E}_{x \sim p_{\text{data}}}[\log D(x)] - \mathbb{E}_{z \sim p(z)}[\log(1 - D(G(z)))] \tag{1}$$

while the generator is trained to minimize[1]:

$$\mathcal{L}_G = -\mathbb{E}_{z \sim p(z)}[\log D(G(z))] \tag{2}$$

Informally, the generator is trained to *trick* the discriminator into believing that the generated samples $G(z)$ actually come from the target distribution, $p(x)$, while the discriminator is trained to be able to distinguish the samples from each other.

**Inception Score and Frechet Inception Distance:** We will refer frequently to the Frechet Inception Distance (FID) (Heusel et al., 2017), to measure the effectiveness of an image synthesis model. To compute this distance, one assumes that we have a pre-trained Inception classifier. One further assumes that the activations in the penultimate layer of this classifier come from a multivariate Gaussian. If the activations on the real data are $N(m, C)$ and the activations on the fake data are $N(m_w, C_w)$, the FID is defined as:

$$\|m - m_w\|_2^2 + \text{Tr}\big(C + C_w - 2\big(CC_w\big)^{1/2}\big) \tag{3}$$

**Core-set selection:** In computational geometry, a Core-set, $Q$, of a set $P$ is a subset $Q \subset P$ that approximates the 'shape' of $P$ (Agarwal et al., 2005). The problem of approximating the shape of a larger set using a smaller subset is formally called shape fitting Lempitsky & Boykov (2007). Core-sets perform sequential selection of points for shape-fitting (Agarwal et al., 2005). Core-sets are used to quickly generate approximate solutions to problems whose full solution on the original set would be burdensome to compute. Given such a problem[2], one computes $Q$, then computes the solution to the problem for $Q$ and converts that into an approximate solution for the original set $P$. The general Core-set selection problem can be formulated several ways, here we consider the the *minimax facility location* formulation (Farahani & Hekmatfar, 2009):

$$\min_{Q:|Q|=k} \max_{x_i \in P} \min_{x_j \in Q} d(x_i, x_j) \tag{4}$$

where $k$ is the desired size of $Q$, and $d(.,.)$ is a metric on $P$. Informally, the formula above encodes the following objective: find some set, $Q$, of points of size $k$ such that the maximum distance between a point in $P$ and its nearest point in $Q$ is minimized. Since finding the exact solution to the minimax facility location problem is NP-Hard (Wolsey & Nemhauser, 2014), we will have to make do with a greedy approximation, detailed in Section 3.3. Using the *minimax facility location* formulation is an appropriate heuristic since Agarwal et al. (2005) demonstrated that minimizing the maximum distance between $Q$ and $P$ is an effective approximate solution for shape fitting.

---

[1] This is the commonly used "Non-Saturating Cost". There are many others, but for brevity and since our technique we describe is agnostic to the loss function, we will omit them.

[2] As an example, consider computing the diameter of a point-set (Agarwal et al., 2005).

---

**Algorithm 1** `GreedyCoreset`

---
**Input:** batch size ($k$), data points ($x$ where $|x| > k$)
**Output:** subset of $x$ of size $k$
    $s \leftarrow \{\}$                                                   ▷ Initialize the sampled set
    **while** $|s| < k$ **do**                           ▷ Iteratively add points to sampled set
        $p \leftarrow \arg\max_{x_i \notin s} \min_{x_j \in s} d(x_i, x_j)$
        $s \leftarrow s \cup \{p\}$
    **end while return** $s$

---

# 3 USING CORE-SET SAMPLING FOR GANs (OR SMALL-GAN)

We aim to use Core-set sampling to increase the effective batch size during GAN training. This involves replacing the basic sampling operation that is done implicitly when minibatches are created. This implicit sampling operation happens in two places: First, when we create a minibatch of samples drawn from the prior distribution $p(z)$. Second, when we create a minibatch of samples from the target distribution $p_{\mathtt{data}}(x)$ to update the parameters of the discriminator. The first of these replacements is relatively simple, while the second presents challenges. In both cases, we have to work around the fact that actually doing Core-set sampling is computationally hard.

## 3.1 SAMPLING FROM THE PRIOR

We need to sample from the prior when we update the discriminator and generator parameters. Our Core-set sampling algorithm doesn't take into account the geometry of the space we sample from, so sampling from a complicated density may cause trouble. This problem is not intractable, but it's nicer not to have to deal with it, so in the absence of any evidence that the choice of prior affects the training, we define the prior in our experiments to be a uniform distribution over a hypercube. To add Core-set sampling to the prior distribution, we randomly sample $n$ points from the prior, where $n$ is greater than the desired batch size, $k$. We then perform Core-set selection on the large batch of size $n$ to create a batch of size $k$. By applying Core-set sampling on the randomly over-sampled prior, we obtain a small sparse batch that approximates the shape of the hypercube. The smaller batch is what's actually used to perform an SGD step.

## 3.2 SAMPLING FROM THE TARGET DISTRIBUTION

Sampling from the target distribution is more challenging. The elements drawn from the distribution are high dimensional images, so taking pairwise distances between them will tend to work poorly due to concentration of distances (Donoho et al., 2000; Sinha et al., 2019), and the fact that Euclidean distances are semantically meaningless in image space (Girod, 1993; Eskicioglu & Fisher, 1995).

To avoid these issues, we instead pre-process our data set by computing the 'Inception Embeddings' of each image using a pre-trained classifier (Szegedy et al., 2017). This is commonly done in the transfer-learning literature, where it is generally accepted that these embeddings have nontrivial semantics (Yosinski et al., 2014). Since this pre-processing happens only once at the beginning of training, it doesn't affect the per-training-step performance.

In order to further reduce the time taken by the Core-set selection procedure, and inspired by the Johnson-Lindenstrauss Lemma (Dasgupta & Gupta, 2003), we take random low dimensional projections of the Inception Embeddings. In practice, we multiply the Inception Embeddings by a fixed and random matrix, to find low-dimensional embeddings where the distances are *nearly* preserved (Dasgupta & Gupta, 2003). Combined with Core-set selection, this gives us low-dimensional representations of the training set images in which pairwise Euclidean distances have meaningful semantics. We can then use Core-set sampling on those representations to select images at training time, analogous to how we select images from the prior.

## 3.3 GREEDY CORE-SET SELECTION

In the above sections, we have invoked Core-set selection while glossing over the detail that exactly solving the $k$-center problem is NP-hard. This is important, because we propose to use Core-set

selection at *every* training step[3]. Fortunately, we can make do with an approximate solution, which is faster to compute: we use the greedy $k$-center algorithm (similar to Sener & Savarese (2017)) summarized in Alg. 1.

## 3.4 SMALL-GAN

Our full proposed algorithm for GAN training is presented in Alg. 2. Our technique is agnostic to the underlying GAN framework and therefore can replace random sampling of mini-batches for all GAN variants. More implementation details and design choices are presented in Section 4.

---

**Algorithm 2** `Small-GAN`

---

**Input:** target batch size ($k$), starting batch size ($n > k$), Inception embeddings ($\phi_I$)
**Output:** a trained GAN
  Initialize networks $G$ and $D$
  **for** $step = 1$ to ... **do**
    $z \sim p(z)$                                    $\triangleright$ Sample $n$ points from the prior
    $x \sim p(x)$                         $\triangleright$ Sample $n$ points from the data distribution
    $\phi(x) \leftarrow \phi_I(x)$                    $\triangleright$ Get cached embeddings for $x$
    $\hat{z} \leftarrow \texttt{GreedyCoreset}(z)$           $\triangleright$ Get Core-set of $z$
    $\widehat{\phi(x)} \leftarrow \texttt{GreedyCoreset}(\phi(x))$      $\triangleright$ Get Core-set of embeddings
    $\hat{x} \leftarrow \phi_I^{-1}(\widehat{\phi(x)})$       $\triangleright$ Get $x$ corresponding to sampled embeddings
    Update GAN parameters as usual
  **end for**

---

# 4 EXPERIMENTS

In this section we look at the performance of our proposed sampling method on various tasks: In the first experiment, we train a GAN on a Gaussian mixture dataset with a large number of modes and confirm our method substantially mitigates 'mode-dropping'. In the second, we apply our technique to GAN-based anomaly detection (Kumar et al., 2019) and significantly improve on prior results. Finally, we test our method on standard image synthesis benchmarks and confirm that our technique seriously reduces the need for large mini-batches in GAN training. The variety of settings in these experiments testifies to the generality of our proposed technique.

## 4.1 IMPLEMENTATION DETAILS

For our Core-set algorithm, the distance function, $d(\cdot, \cdot)$ s the $\ell_2$-norm for both the prior and target distributions. Since we use the current state-of-the-art GAN architectures, each GAN model is trained with the same hyper-parameters and optimizers as was proposed in the corresponding paper, to ensure fair comparison. The only hyper-parameter altered is the batch-size, which is stated for all experiments. For over-sampling, we use a factor of 4 for the prior $p(z)$ and a factor of 8 for the target, $p(x)$, unless otherwise stated. We investigate the effects of different over-sampling factors in the ablation study in Section 4.6.

## 4.2 MIXTURE OF GAUSSIANS

We first investigate the problem of mode dropping (Arora et al., 2018) in GANs, where the GAN generator is unable to recover some modes from the target data set. We investigate the performance of training a GAN to recover a different number of modes of 2D isotropic Gaussian distributions, with a standard deviation of 0.05. We use a similar experimental setup as Azadi et al. (2018), where our generator and discriminator are parameterized using 4 ReLU-fully connected networks, and use the standard GAN loss in Eq. 1 and 2. To evaluate the performance of the models, we generate

---

[3] Though the Core-set sampling does happens on CPU and so could be done in parallel to the GPU operations used to train the model, as long as the Core-set sampling time doesn't exceed the time of a forward and backward pass – which it doesn't.

| Number of Modes | % of Recovered Modes (GAN) | % of Recovered Modes (Ours) | % of High-Quality Samples (GAN) | % of High-Quality Samples (Ours) |
|---|---|---|---|---|
| 25 | **100** | **100** | 95.76 | **98.9** |
| 36 | **100** | **100** | 92.73 | **95.34** |
| 49 | 98.12 | **99.85** | 84.28 | **88.1** |
| 64 | 96.13 | **99.01** | 68.81 | **82.11** |
| 81 | 92.59 | **98.84** | 49.74 | **71.75** |
| 100 | 90.67 | **97.33** | 23.31 | **49.87** |

Table 1: Experiments with large number of modes

| Held-out Digit | Bi-GAN | MEG | Core-set+MEG |
|---|---|---|---|
| 1 | 0.287 | 0.281 | **0.351** |
| 4 | 0.443 | 0.401 | **0.501** |
| 5 | 0.514 | 0.402 | **0.518** |
| 7 | 0.347 | 0.29 | **0.387** |
| 9 | 0.307 | 0.342 | **0.39** |

Table 2: Experiments with Anomaly Detection on MNIST dataset. The Held-out digit represents the digit that was held out of the training set during training and treated as the anomaly class. The numbers reported is the area under the precision-recall curve.

$10,000$ samples and assign them to their closest mode. As in Azadi et al. (2018), the metrics we use to evaluate performance are: $i$) 'high quality samples', which are samples within 4 standard deviations of the assigned mode and $ii$) 'recovered modes' which are mixture components with at least one assigned sample.

Our results are present in table 1, where we experiment with an increasing number of modes. We see that as the number of modes increases, a normal GAN suffers from increased mode dropping and lower sample quality compared to Core-set selection. With 100 modes, Core-set selection recovers 97.33% of the modes compared to 90.67% for the vanilla GAN. Core-set selection also generates 49.87% 'high quality' samples compared to 23.31% for the vanilla GAN.

### 4.3 ANOMALY DETECTION

To see whether our method can be useful for more than just GANs, we also apply it to the Maximum Entropy Generator (MEG) from Kumar et al. (2019). MEG is an energy-based model whose training procedure requires maximizing the entropy of the samples generated from the model. Since MEG gives density estimates for arbitrary data points, it can be used for anomaly detection – a fundamental goal of machine learning research (Chandola et al., 2009; Kwon et al., 2017) – in which one aims to find samples that are 'atypical' given a source data set. Kumar et al. (2019) do use MEG successfully for this purpose, achieving results close to the state-of-the-art technique for GAN-based anomaly detection (Zenati et al., 2018). We hypothesized that – since energy estimates can in theory be improved by larger batch sizes – these results could be further improved by using Core-set selection, and we ran an experiment to confirm this hypothesis.

We follow the experimental set-up from Kumar et al. (2019) by training the MEG with all samples from a chosen MNIST digit left-out during training. Those samples then serve as the 'anomaly class' during evaluation. We report the area under the precision-recall curve and average the score over the last 10 epochs. The results are reported in Table 2, which provides clear evidence in favor of our above hypothesis: for all digits tested, adding Core-set selection to MEG substantially improves the results. By performing these experiments, we aim to show the general applicability of Core-set selection, not to suggest that MEG is superior to BiGANs (Zenati et al., 2018) on the task. We think it's likely that similar improvements could be achieved by using Core-set selection with BiGANs.

| GAN (batch-size = 128) | Small-GAN (batch-size = 128) | GAN (batch-size = 256) | Small-GAN (batch-size = 256) | GAN (batch-size = 512) | Small-GAN (batch-size = 512) |
|---|---|---|---|---|---|
| $18.75 \pm 0.2$ | $\mathbf{16.73 \pm 0.1}$ | $17.9 \pm 0.1$ | $\mathbf{16.22 \pm 0.3}$ | $15.68 \pm 0.2$ | $\mathbf{15.08 \pm 0.1}$ |

Table 3: FID scores for CIFAR using SN-GAN as the batch-size is progressively doubled. The FID score is calculated using $50,000$ generated samples from the generator.

| Small-GAN (batch-size = 64) | GAN (batch-size = 64) | GAN (batch-size = 128) | GAN (batch-size = 256) |
|---|---|---|---|
| 13.08 | 14.82 | 13.02 | 12.63 |

Table 4: FID scores for LSUN using SAGAN as the batch-size is progressively doubled. The FID score is calculated using $50,000$ generated samples from the generator. All experiments were run on the 'outdoor church' subset of the dataset.

## 4.4 IMAGE SYNTHESIS

**CIFAR and LSUN:** We also conduct experiments on standard image synthesis benchmarks. To further show the generality of our method, we experiment with two different GAN architectures and two image datasets. We use Spectral Normalization-GAN (Miyato et al., 2018) and Self Attention-GAN (Zhang et al., 2018) on the CIFAR (Krizhevsky et al., 2009) and LSUN (Yu et al., 2015) datasets, respectively. For the LSUN dataset, which consists of 10 different categories, we train the model using the 'outdoor church' subset of the data.

For evaluation, we measured the FID scores (Heusel et al., 2017) of $50,000$ generated samples from the trained models[4]. We compare the performance using SN-GANs with and without Core-set selection across progressively doubling batch sizes. We observe a similar effect to Brock et al. (2018): just by increasing the mini-batch size by a factor of 4, from 128 to 512, we are able to improve the FID scores from 18.75 to 15.68 for SN-GANs. This further demonstrates the importance of large mini-batches for GAN training. Adding Core-set selection significantly improves the performance of the underlying GAN for all batch-sizes. For a batch size of 128, our model using Core-set sampling significantly outperforms the normal SN-GAN trained with a batch size of 256, and is comparable to an SN-GAN trained with a batch size of 512. The results suggest that the models perform significantly better for any given batch size when Coreset-sampling is used.

However, Core-set sampling does become less helpful as the underlying batch size increases: for SN-GAN, the performance improvement at a batch size of 128 is much larger than the improvement at a batch size of 512. This supports the hypothesis that Core-set selection works by approximating the coverage of a larger batch; a larger batch can already recover more modes of the data - so under this hypothesis, we would expect Core-set selection to help less.

We see similar results when experimenting with Self Attention GANs (SAGAN) (Zhang et al., 2018) on the LSUN dataset (Yu et al., 2015). Compared to our results with SN-GAN, increasing the batch size results in a smaller difference in the performance for the SAGAN model, but we still see the FID improve from 14.82 to 12.63 as the batch-size is increased by a factor of 4. Using Core-set sampling with a batch size of 64, we are able to achieve a comparable score to when the model is trained with a batch size of 128. We believe that one reason for a comparably smaller advantage of using Core-set sampling on LSUN is the nature of the data itself: using the 'outdoor church' subset of LSUN reduces the total number of 'modes' *possible* in the target distribution, since images of churches have fewer differences than the images in CIFAR-10 data set. We see similar effects in the mixture of Gaussians experiment (See 4.2) where the relative difference between a GAN trained with and without Core-set sampling increases as the number of modes are increased.

**ImageNet:** Finally, in order to test that our method would work 'at scale, we ran an experiment on the ImageNet data set. Using the code at `https://github.com/heykeetae/Self-Attention-GAN`, we trained two GANs: The first is trained exactly as described in the

---

[4]Note that we measure the performance of all the models using the PyTorch version of FID scores, and not the official Tensorflow one. We ran all our experiments with the same code for accurate comparison.

| Small-GAN (batch size = 128) | SN-GAN (batch size = 128) | SN-GAN (batch size = 256) | SN-GAN (batch size = 512) |
|---|---|---|---|
| 14.51 | 13.31 | 26.46 | 51.64 |

Table 5: Timing to perform 50 gradient updates for SN-GAN with and without Core-sets. The time is measured in seconds. All the experiments were performed on a single NVIDIA Titan-XP GPU. The sampling factor was 4 for the prior and 8 for the target distribution.

| Small-GAN | A | B | C | D | E |
|---|---|---|---|---|---|
| **16.73** | 18.75 | 18.09 | 17.03 | 17.88 | 17.45 |

Table 6: FID scores for CIFAR using SN-GAN. The experiment list is: A = Training an SN-GAN, B = Core-set selection directly on the images, C = Core-set applied directly on Inception embeddings without a random projection, D = Core-set applied only on the prior distribution, E = Core-set applied only on target distribution.

open-source code. The second is trained using Coreset selection, with all other hyper-parameters unchanged. Simply adding Coreset selection to the existing SAGAN code materially improved the FID (which we compute using 50000 samples): the baseline model had an FID of 19.40 and the Core-set model had an FID of 17.33.

## 4.5 TIMING ANALYSIS

Since random sampling can be done very quickly, it is important to investigate the amount of time it takes to train GANs with and without Core-set sampling. We measured the time for SN-GAN to do 50 gradient steps on the CIFAR dataset with various mini-batch sizes: the results are in Table 5. On average, for each gradient step, the time added by performing Core-Set sampling is only 0.024 seconds.

## 4.6 ABLATION STUDY

We conduct an ablation study to investigate the reasons for the effectiveness of Core-set selection. We also investigate the effect of different sampling factors and other hyper-parameters. We run all ablation experiments on the task of image synthesis using SN-GAN (Miyato et al., 2018) with the CIFAR-10 dataset (Krizhevsky et al., 2009). We use the same hyperparameters as in our main image synthesis experiments and a batch size of 128, unless otherwise stated.

## 4.7 EXAMINATION OF MAIN HYPER-PARAMETERS

We examine $i)$ the importance of the chosen target distribution for Core-set selection and $ii)$ the importance of performing Core-set on that target distribution. The FID scores are reported in Table 6. The importance of the target distribution is clear, since performing Core-set selection directly on the images (experiment B) performs similar to random-sampling. Experiment C supports our hypothesis that performing a random projection on the Inception embeddings can preserve semantic information while reducing the dimensionality of the features. This increases the effectiveness of Core-set sampling and reduces sampling time. Our ablation study also shows the importance of performing Core-set selection on both the prior and target distribution. The FID scores of the models are considerably worse when Core-set sampling is used on either distribution alone.

## 4.8 EXAMINATION OF SAMPLING FACTORS

Another important hyper-parameter for training GANs using Core-set selection is the sampling factor. In Table 7 we varied the factors by which both the prior and the target distributions were over-sampled. We see that using 4 for the sampling factor for the prior and 8 for the sampling factor for the target distribution results in the best performance. One interesting observation: that the performance eventually starts degrading as the sampling factor is increased. Since the greedy algorithm in Alg. 1 sequentially selects the point that is the furthest away from the the already sampled set, we

| A | B | C | D | E | F | G | H | I |
|---|---|---|---|---|---|---|---|---|
| 18.01 | 17.8 | 17.59 | 17.12 | 16.83 | **16.73** | 16.9 | 17.95 | 20.79 |

Table 7: FID scores for CIFAR using SN-GAN. Each of the experiment shows a different pair of over-sampling factors for the prior and target distributions. The factors are listed as: sampling factor for prior distribution $\times$ sampling factor for target distribution. A = $2 \times 2$; B = $2 \times 4$; C = $4 \times 2$; D = $4 \times 4$; E = $8 \times 4$; F = $4 \times 8$; G = $8 \times 8$; H = $16 \times 16$; I = $32 \times 32$

believe that when the sampling factors are set too high, the algorithm becomes sensitive to outliers. See the appendix for more discussion of this phenomenon.

## 5 RELATED WORK

**Variance Reduction in GANs:** Researchers have proposed reducing variance in GAN training from an optimization perspective, by directly changing the way each of the networks are optimized. Some have proposed applying the extragradient method (Chavdarova et al., 2019), and others have proposed casting the *minimax* two-player game as a variational-inequality problem (Gidel et al., 2018). Brock et al. (2018) recently proposed reducing variance by using large mini-batches.

**Stability in GAN Training:** Stabilizing GANs has been extensively studied theoretically. Researchers have worked on improving the dynamics of the two player minimax game in a variety of ways (Nagarajan & Kolter, 2017; Mescheder et al., 2018; Mescheder, 2018; Li et al., 2017b; Arora et al., 2017). Training instability has been linked to the architectural properties of GANs: especially to the discriminator (Miyato et al., 2018). Proposed architectural stabilization techniques include using Convolutional Neural Networks (CNNs) (Radford et al., 2015), using very large batch sizes (Brock et al., 2018), using an ensemble of the discriminators (Durugkar et al., 2016), using spectral normalization for the discriminator (Miyato et al., 2018), adding self-attention layers for the generator and discriminator networks (Vaswani et al., 2017; Zhang et al., 2018) and using iterative updates to a *global* generator and discriminator using an ensemble(Chavdarova & Fleuret, 2018). Different objectives have also been proposed to stabilize GAN training (Arjovsky et al., 2017; Gulrajani et al., 2017; Li et al., 2017a; Mao et al., 2017; Mroueh & Sercu, 2017; Bellemare et al., 2017).

**Core-set Selection:** Core-set sampling has been widely studied from an algorithmic perspective in attempts to find better approximate solutions to the original NP-Hard problem (Agarwal et al., 2005; Clarkson, 2010; Pratap & Sen, 2018). The optimality of the sub-sampled solutions have also been studied theoretically (Barahona & Chudak, 2005; Goldman, 1971). See Phillips (2016) for a recent survey on Core-set selection algorithms. Core-sets have been applied to many machine learning problems such as $k$-means and approximate clustering (Har-Peled & Mazumdar, 2004; Har-Peled & Kushal, 2007; Bādoiu et al., 2002)), active learning for SVMs (Tsang et al., 2005; 2007), unsupervised subset selection for hidden Markov models (Wei et al., 2013) scalable Bayesian inference, (Huggins et al., 2016) and mixture models (Feldman et al., 2011). We are not aware of Core-set selection being applied to GANs.

**Core-set Selection in Deep Learning:** Core-set selection is largely underexplored in the Deep Learning literature, but interest has recently increased. Sener & Savarese (2017) proposed to use Core-set sampling as a batch-mode active learning sampler for CNNs. Their method used the embeddings of a trained network to sample from. Mussay et al. (2019) proposed using Core-set selection on the activations of a neural network for network compression. Core-set selection has also been used in continual learning to sample points for episodic memory (Nguyen et al., 2017).

## 6 CONCLUSION

In this work we present a general way to mimic using a large batch-size in GANs while minimizing computational overhead. This technique uses Core-set selection and improves performance in a wide variety of contexts. This work also suggests further research: a similar method could be applied to other learning tasks where large mini-batches may be useful.

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

# A APPENDIX

## A.1 POTENTIAL DISTORTING AFFECTS OF CORE-SET SAMPLING

Theoretically, Core-set sampling of the training distribution can cause some distortion of the distribution learned by the generator. However, there are good ways to mitigate this, as we will discuss. Moreover, in cases where Core-set sampling causes more distortion, a baseline GAN will tend to distort things in the opposite direction.

In more detail: We train GANs using 4 different algorithms on a mixture of two Gaussians with mixing coefficients of 0.01 and 0.99. For all 4 experiments, we use a batch size of 10. The results are as follows:

**Vanilla GAN:** Training a 'vanilla' GAN on this mixture of two Gaussians results in the 0.01-mode being completely dropped. This is consistent with existing results on GANs. It's also important because it suggests that, even if Core-set sampling is unavoidably distorting (which it's not - see below), there might not be a non-distorting baseline that works in the same situation.

Core-set Sampling Training a GAN with Core-set sampling (sampling factor of 20) results in considerable distortion: 90.9% of samples are from the 0.99-mode and 8.9% of samples are from the 0.01-mode (the rest of the samples were over 4 standard deviations from either mean) - so the 0.01 mode is over-represented by about a factor of 10. This makes a lot of sense: the final batch size is 10, and we take 200 samples to start, so the expected number of 0.01-mode samples in our original batch is 2. Core-set sampling will ensure we keep at least 1 of those 2 in our final batch of 10, yielding a 10% representation for a mode that should only have 1% representation. It's worth pointing out that you have to make the skew in mixture components quite high relative to the batch size for this to matter, but there likely exist interesting data-sets that have this characteristic.

Thus, we test two methods for reducing the distortion:

**Anneal Sampling Factors:** Training a GAN with Core-set sampling but annealing the sampling factor from 20 to 2 during training results in 98.3% and 1.6% of samples from the 0.99 and 0.01 modes, respectively. This is a substantial reduction in distortion.

**Fine-Tune on Randomly Sampled Data:** Training a GAN with Core-set sampling to start and then random sampling near the end results in 99.5% and 0.5% of samples from the 0.99 and 0.01 modes, respectively, for our best try. This method resulted in some instability - the longer the GAN is training randomly, the more likely the GAN was to drop the 0.01 mode. For this reason we would probably recommend the annealing method in practice.

**Final Observations:** We have two more observations to make about this: First, the FID should in principle be sensitive to such distortions, and so the fact Core-set sampling was able to improve the FID in some sense suggests that – on CIFAR, LSUN, and ImageNet — the distortion was not that significant. Second, Azadi et al. (2018) seems like another promising way to reduce distortion, but we did not test it since the annealing method described above seems satisfactory.

