# OpenReview forum: "Small-GAN: Speeding up GAN Training using Core-Sets"
_ICLR.cc/2020/Conference — Reject_

### Official Review · AnonReviewer3 · 2019-10-22
**Official Blind Review #3**

**Rating:** 6

**Review:**

Summary:
This paper addresses the challenging problem of how to speed up the training of GANs without using large mini-batch sizes and causing significant performance drop. To achieve this, the authors propose to use the method of core-sets, mainly inspired by recent use of core-set selection in active learning. The proposed method allows us to generate effectively large mini-batches though actually small during the training process, or more concretely, drawing a large batch of samples from the prior and then compress that batch using core-set selection. To address the curse of dimensionality issue for high-dimensional data like images, the authors suggest using a low-dimensional embedding based on Inception activations of each training image. Regarding the experimental evaluation, it is clearly shown that the proposed core-set selection greatly improves GAN training in terms of timing and memory usage, and allows significantly reducing mode collapse on a synthetic dataset. As a by-product, it is successfully applied to anomaly detection and achieves state-of-the-art results.

Strengths:
The paper is generally well written and clearly presented.  As mentioned in the text, the use of core-sets is not novel in machine learning, but unfortunately not yet sufficiently explored in deep learning, and there are still few useful tools available in the literature. I believe this work will have a positive impact on the community and especially help establishing more efficient methods for training GANs.

Weaknesses:
- Experimental results are indeed very promising, however, GAN implementation details and hyperparameters used for training, such as optimizer and learning rate, do not seem to be mentioned in the text. I think this would be helpful for readers to better understand how this all works.
- There does not seem to be any discussion on the convergence and stability of GAN training, which should be clarified in the experimental section.
- On page 3, in Sect. 3.2,  I find “random low dimensional projections of the Inception Embeddings” is not clear, more technical details should be provided.

**Experience Assessment:**

I have read many papers in this area.

**Review Assessment: Checking Correctness Of Derivations And Theory:**

I assessed the sensibility of the derivations and theory.

**Review Assessment: Checking Correctness Of Experiments:**

I assessed the sensibility of the experiments.

**Review Assessment: Thoroughness In Paper Reading:**

I read the paper at least twice and used my best judgement in assessing the paper.

---

> ### Author Response · Authors · 2019-11-14
> **Thank you for the review!**
>
> Thank you for the review!
>
> We have added details about the experiments, and the hyperparameters used, as well as clarified some details in the text, as suggested by the review. We have fixed expanded on what we mean by "random low dimensional projections", and how we do that in practice. We also added results with the ImageNet dataset in the draft.

---

### Official Review · AnonReviewer1 · 2019-10-22
**Official Blind Review #1**

**Rating:** 3

**Review:**

Training with large batches provides a disproportionate improvement for GANs (e.g. FID drops from 18.65 to 12.39 by simply increasing the batch size by a factor of 8 for BigGAN). The authors point out that not everybody has access to the computing power which is required to run large batches. Therefore, this paper proposes a method to select the image of the batch and thereby obtaining the benefits of large batches while running only small batches.

The authors perform coreset selection of a large set of images (actually a greedy variant from Sener & Savarese). The selected images are then used for training with small batch size. The coreset selection is applied to the inception activations of the images (or a randomly downsampled version of them). Experiments show that the trained GANs obtain better mode coverage, improve anomaly detection,  and obtain GANs with higher FID scores on image synthesis.

The paper is well motivated and solutions that prevent having to use large batches will have a significant impact on the field.

Conclusion: At the moment I recommend a weak reject. The technical contribution of the paper is rather small, and also the depth of analysis could be improved. Moreover, I think experimental results could have been more complete. However, given the importance of the problem addressed in the paper (and the little existing work) I could be open to increasing my score if my concerns are addressed.

Main points:
1. GANs aim to generate high-quality images. In addition, they aim to generate these high-quality images according to the distribution of the train dataset. The main danger of not randomly sampling from the train distribution is that the trained GAN does no longer generate images according to the train distribution. I think this issue should be more clearly discussed and evaluated in the paper (for some applications this might not be a big problem for others yes). For example, if you would introduce mixing coefficients for the Gaussian mixture and Gaussians with different variances (section 4.2), it could be possible that the GAN would generate according to these mixing coefficients with more accuracy than small-GAN (because the coreset selection would introduce a bias). The proposed metrics do not measure this.
Maybe this could simply be solved by adding some final epochs which just sample randomly again?
2. A naive approach to creating large batches is by only updating the network every N batches, and summing the gradients of the N batches. I would like to see this option discussed and compared to. I can see how the BN is different from a large batch but it would still be interesting to see.
3. I think the examination of sampling factors should be explained in more detail. The importance of the sampling factor of the prior is harder to understand (and might be necessary to counter the effect I discuss in 1?) Is it not expected that both these sampling factors should be equal?


Minor points (do not need to be addressed in rebuttal):
-Since FID several other GAN evaluation metrics have been proposed. I think the authors could also consider ‘Assessing Generative Models via Precision and Recall’ or ‘Improved precision and recall metric for assessing generative models’ for a more complete insight insight
-I am not convinced this GAN needs a name (small-GAN), especially since it can be applied to other GAN architectures.
-The claim in the abstract for state of the art in anomaly detection should be removed (no extensive study nor comparison is performed)
-I prefer to see the venue of publication when possible (for example Sener & Savarese is ICLR 2018)


**Experience Assessment:**

I have read many papers in this area.

**Review Assessment: Checking Correctness Of Derivations And Theory:**

I assessed the sensibility of the derivations and theory.

**Review Assessment: Checking Correctness Of Experiments:**

I assessed the sensibility of the experiments.

**Review Assessment: Thoroughness In Paper Reading:**

I read the paper at least twice and used my best judgement in assessing the paper.

---

> ### Author Response · Authors · 2019-11-11
> **Thanks for the review!**
>
> Thanks for the review!
>
> We respond to your 3 main points in separate comments for clarity:
> We are updating the draft now to correspond to all these points as well, but these comments will likely be easier parse than diffs to the PDF.
> We hope that these improvements merit some increase in score.

---

> ### Author Response · Authors · 2019-11-11
> **Response to Point 1  (On Distortion)**
>
> This is a great point and something we didn't give enough attention to originally.
> We have run an extra set of experiments (which we will describe below) to address this.
>
> Broadly speaking, Core-set sampling of the training distribution can cause the type of distortion you described,
> but there are also good ways to mitigate it, as you suggest.
> Moreover, in cases where Core-set sampling causes more distortion, a baseline GAN will tend to distort things in the opposite direction.
>
> In more detail:
> We train GANs using 4 different algorithms on a mixture of two Gaussians with mixing coefficients of 0.01 and 0.99.
> For all 4 experiments, we use a batch size of 10.
> The results are:
>
> a) Training a `vanilla' GAN on this mixture of two Gaussians results in the 0.01-mode being completely dropped.
> This is consistent with existing results on GANs.
> It's also important because it suggests that, even if Core-set sampling is unavoidably distorting (which it's not - see below),
> there might not be a non-distorting baseline that works in the same situation.
>
> b) Training a GAN with Core-set sampling (sampling factor of 20) results in exactly the type of distortion you predicted: 90.9% of samples are from the 0.99-mode
> and 8.9% of samples are from the 0.01-mode (the rest of the samples were over 4 standard deviations from either mean)
> - so the 0.01 mode is over-represented by about a factor of 10.
> This makes a lot of sense: the final batch size is 10, and we take 200 samples to start, so the expected number of 0.01-mode samples in our
> original batch is 2. Core-set sampling will ensure we keep at least 1 of those 2 in our final batch of 10, yielding a 10% representation
> for a mode that should only have 1% representation.
> It's worth pointing out that you have to make the skew in mixture components quite high relative to the batch size for this to matter,
> but we agree that there likely exist interesting data-sets that have this characteristic.
>
> Thus, we test two methods for reducing the distortion:
>
> c) Training a GAN with Core-set sampling but annealing the sampling factor from 20 to 2 during training results in
> 98.3% and 1.6% of samples from the 0.99 and 0.01 modes, respectively.
> This is a substantial reduction in distortion.
>
> d) Training a GAN with Core-set sampling to start and then random sampling near the end (as you suggest) results in
> 99.5% and 0.5% of samples from the 0.99 and 0.01 modes, respectively, for our best try.
> This method resulted in some instability - the longer the GAN is training randomly, the more likely the GAN was to drop the 0.01 mode.
> For this reason we would probably recommend method c) in practice.
>
> We have two more observations to make about this.
> First, the FID should in principle be sensitive to such distortions, and so the fact Core-set sampling was able to improve the FID
> in some sense suggests that -- on CIFAR, LSUN, and ImageNet --- the distortion was not that significant.
> Second, Discriminator Rejection Sampling [1] seems like another promising way to reduce distortion, but
> we did not test it since method c) above seems pretty satisfactory.
>
> [1] Discriminator Rejection Sampling (Azadi et al)

---

> ### Author Response · Authors · 2019-11-11
> **Response to Point 2 (On Gradient Accumulation)**
>
> We do mention gradient accumulation a bit in the third paragraph, but we will expand on this.
> With the exception of batch-wise operations (like batch norm, as you noted), it should be semantically identical to using large batches.
>
> We ran a small experiment just to check:
> With SN-GAN on CIFAR10 and a 'true' batch size of 128, we get the following results:
>
> 1 gradient accumulation  (128 effective batch size): 18.75 +/- 0.2
>
> 2 gradient accumulations (256 effective batch size): 17.995 +/- 0.2
>
> 4 gradient accumulations (512 effective batch size): 15.83 +/- 0.2
>
> Recall from our Table 3 that the numbers from training SNGANS with actual
> batches of size 256 and 512 are 17.9 and 15.68, respectively.
> These numbers are very close to those numbers, suggesting that e.g. the difference
> in batch norm was not significant.

---

> ### Author Response · Authors · 2019-11-11
> **Response to Point 3 (On Sampling Factors)**
>
> We will expand the section on 'Examination of Sampling Factors' in the draft.
> > Is it not expected that both these sampling factors should be equal?
> Two thoughts on this:
> First, see this reproduction of part of table 7:
>
> Prior Factor   |  Target Factor |    FID
> --------------------------------------------------------------
>          8             |            4             |   16.83
>          4             |            8             |   16.73
>          8             |            8             |   16.9
>
> We end up using the middle configuration in other experiments because
> it was best, but you can see that the FID difference between the configuration
> we used and the 8 X 8 configuration is only around 1.5 standard deviations
> (According to the FID numbers from Table 3).
>
> Second, it may be that the distortion phenomenon you describe in point 1
> (which won't really apply to the prior) could cause different factors to be optimal
> for the prior and the target distributions, but if so the difference doesn't seem
> particularly significant.

---

> ### Author Response · Authors · 2019-11-14
> **Updated Impression?**
>
> Hello,
>
> We believe we have addressed your concerns and clarified some points you raised. Do you have an updated impression of our paper? Should that not be the case, please do not hesitate to get in touch with us. Thanks for your consideration and time. We appreciate it!

---

### Official Review · AnonReviewer2 · 2019-10-25
**Official Blind Review #2**

**Rating:** 6

**Review:**

This paper applies core-set selection to the training of GANs. The motivation is to limit the minibatch size with suitably sampled sets of datapoints. The proposed technique is relatively reasonable: e.g. extract features from an image, reduce dimensionality by the taking random projections, then run Core-Set selection. The Core-Set selection part of the method is modular from the rest of the GAN training, and can be applied easily.

Generally, I think this is reasonabl work. While the idea itself is not extremely novel, it is interesting to see CoreSets applied to GANs. The paper can be made stronger if there is more discussion about the theory of CoreSets and how good are the heuristics used in this paper.





**Experience Assessment:**

I do not know much about this area.

**Review Assessment: Checking Correctness Of Derivations And Theory:**

N/A

**Review Assessment: Checking Correctness Of Experiments:**

I assessed the sensibility of the experiments.

**Review Assessment: Thoroughness In Paper Reading:**

I read the paper at least twice and used my best judgement in assessing the paper.

---

> ### Author Response · Authors · 2019-11-14
> **Thank you for the review!**
>
> Thank you for the review!
>
> In accordance to the suggestions, we have added more detail on the theory of core-sets and comment on the reasonability of the minimax facility location in our case. Specifically, we detail how core-sets solve the problem of shape-fitting, and how prior work shows that the minimax facility location is a good heuristic for shape fitting. We also added results with the ImageNet dataset in the draft.

---

### Author Response · Authors · 2019-10-24
**Experimental Update: Coreset selection improves ImageNet FID for SAGAN**

We have a brief experimental update.
In order to test that our method would work 'at-scale', we ran an experiment on the ImageNet data set.
Using the code at  https://github.com/heykeetae/Self-Attention-GAN, we trained two GANs:
The first is trained exactly as described in the open-source code.
The second is trained using Coreset selection, with all other hyper-parameters unchanged.
Simply adding Coreset selection to the existing SAGAN code materially improved the FID
(which we compute using 50000 samples), as shown here:

SAGAN: 19.40
SAGAN+Coreset: 17.33

We will add these results to the paper once we're able.

---

### Author Response · Authors · 2019-11-14
**Submission update**

We would like to thank each of the reviewers for their time!

We have expanded and clarified the details that the reviewers suggested, and added more about the theory of core-sets. We also expanded on the role of sampling factors, and how they help with the performance of the GAN model. Additionally, we have also added the ImageNet results into the draft, as well as moved the discussion related to the oversampling of the underrepresented modes (as pointed out by R2) into the appendix.

---

### Decision · Program_Chairs · 2019-12-19

**Decision:**

Reject

**Comment:**

The paper proposes to use greedy core set sampling to improve GAN training with large batches. Although the problem is clear and the solution works, reviewers have raised several concerns. One concern is that the technical novelty is limited; another (in the first version) that even a simpler version of gradient accumulation can solve the  main task (rather that computing core-sets). In the end, some discussion was done, with quite a few additions and experiments done by the authors. The final concern that seemingly was not addressed: the gradient accumulation seems to give the same numbers as large batches, thus you can 'mimic' large batch sizes with smaller ones and gradient accumulation, making the main claim of the paper questionable. The achievement of SOTA is good, but it is not clear wether it is due to the proposed technique, or rather smart tuning of a larger set of hyperparameters. Thus, I would agree with the concern of Reviewer1.

---

> ### Author Response · Authors · 2019-12-20
> **Of course gradient accumulation works -- it's just way slower!**
>
> You're missing an important point about gradient accumulation: it's way slower than Core-set sampling.
> There's nothing we're trying to hide here - we mention accumulation right in the introduction.
>
> If you accumulate gradients across N batches with batch size B,
> your training procedure is (roughly) N times slower than if you used one big batch of batch size B*N.
> On the other hand, if you use core-set sampling to create batches of size B from batches of size B*N,
> your training procedure is not really slowed down at all;
> Core-set sampling takes negligible time compared to taking an SGD step, and it could actually be done
> in a separate thread if you wanted.
>
> Regarding hyper-parameter tuning - we don't do tuning, as explained in 4.1.